# IAUNet: Instance-Aware U-Net

## Abstract

Instance segmentation is critical in biomedical imaging for accurately distinguishing individual objects, such as cells, which often overlap and vary in size. Recent query-based methods—where object-specific queries guide segmentation—have shown strong performance in this task. While U-Net has been a go-to architecture in medical image segmentation, it was neither specifically designed for instance segmentation nor explored in the context of query-based approaches. In this work, we present IAUNet, a novel architecture that brings instance awareness to U-Net with query-based mechanisms to achieve superior pixel-to-instance clustering. The key design includes lightweight Instance Activation (IA) layers, which generate guided object queries by highlighting semantically important regions. Additionally, we propose a Parallel Dual-Path Transformer decoder that refines object-specific features across multiple scales, allowing us to assign multiple queries from different scale levels to a specific object. Finally, we introduce the 2025 Revvity Full Cell Segmentation Dataset, comprising hundreds of manually labeled cells from brightfield images. This dataset is unique in capturing the complex morphology of overlapping cell cytoplasm with an unprecedented level of detail, making it a valuable resource and benchmark for advancing instance segmentation in biomedical imaging. Experiments on multiple public datasets and our own show that IAUNet outperforms most state-of-the-art fully convolutional, transformer-based, and query-based models, setting a strong baseline for medical image instance segmentation tasks.

## 1 Introduction

Studying biological systems at the cellular and tissue levels is essential for understanding complex biological processes. At the cellular level, research provides valuable quantitative information on individual cell properties, including shape, position, signaling pathways, and RNA/protein expressions Boutros et al. (2015) Björklund et al. (2006). On the other hand, tissue-level studies reveal collective cell behavior within the context of development and disease. Integrating both approaches leads to a more comprehensive understanding of biological systems, supporting the development of treatments for diseases like cancer, Alzheimer's, and cardiovascular disorders Pös et al. (2018).

Deep learning models have significantly advanced biomedical imaging by outperforming traditional methods and, in some cases, exceeding human expertise He et al. (2015). These models have transformed image segmentation tasks in biomedical imaging, leading to breakthroughs in understanding disease processes and treatment development. Image segmentation using deep learning has become increasingly essential in understanding complex biological structures and processes Liu et al. (2021). Among tasks, cell segmentation – identifying and separating individual cells within images – has become a key area of research. Cell segmentation involves identifying and separating individual cells within images. Deep learning make it possible to obtain quantitative data on cell characteristics, such as shape and position.

However, cell segmentation faces challenges due to the heterogeneity of biological samples. Variations in object count, cell proximity, and overlapping instances make it hard for the models to perform well on segmentation tasks. Among imaging techniques, brightfield microscopy remains popular for its simplicity, low cost, and versatility Morrison et al. (2020) Wang & Fang (2012). It involves emitting natural light through samples and capturing resulting images. Brightfield imaging does not require complex equipment or sample labeling and allows real-time observation of cellular processes. While techniques like fluorescence microscopy require specialized training and equip-

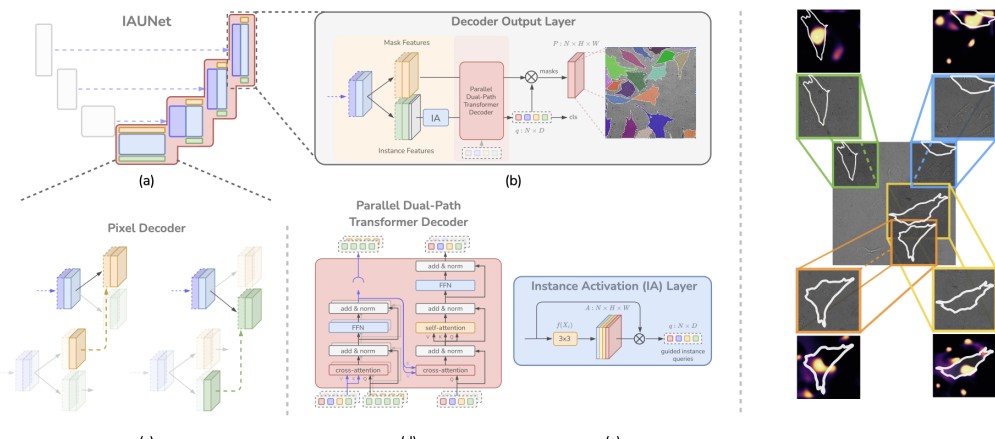

Figure 1: (a) IAUNet model. (b) Parallel Dual-Path Transformer Decoder processes mask and instance features concurrently. (c) Pixel Decoder extracts pixel-wise features. (d) Transformer decoder refines object features across scales. (e) Instance Activation (IA) Layer generates guided instance queries for effective pixel-to-instance clustering.

ment, brightfield microscopy is widely used in biological research and medical diagnostics Ali et al. (2022) Fishman et al. (2021) Salem et al. (2021). Despite its popularity, brightfield image segmentation has received less attention than other modalities due to its complex, noisy, and variable nature, making precise cell segmentation challenging.

Many previous works have been designed specifically for instance segmentation of natural objects and directly applied to the medical imaging domain without making model-specific adjustments. With such a significant domain shift, these methods often underperform when it comes to segmenting individual objects in microscopic samples Follmann & König (2020). In contrast to many approaches, U-Net Ronneberger et al. (2015) has been a go-to method for semantic segmentation due to its robustness and effectiveness in handling complex structures and intricate details Zhou et al. (2018).

Recently, following the success of DETR Carion et al. (2020) in object detection, query-based single-stage instance segmentation methods have gained prominence. These methods move away from traditional convolutional approaches, instead utilizing the powerful attention mechanism Cheng et al. (2022a) together with learnable queries to directly predict object classes and segmentation masks.

In this work, we bridge the gap between the U-Net model, a powerful architecture for biomedical imaging, and the task of instance segmentation, offering fine-grain segmentation that outperforms specialized architectures across various segmentation tasks in the medical domain. We demonstrate that our query-based U-Net variant achieves top-tier performance for instance segmentation task in the medical imaging domain.

Our primary focus is to develop a robust method for cell segmentation in medical images. We extend the U-Net architecture by introducing a novel pixel decoder with decoupled branching on each of its levels, which makes the model instance-aware and capable of adapting to varying object shapes and sizes. Additionally, we integrate a Transformer decoder to enhance the model's ability to capture rich semantic features. Within the Transformer decoder, we employ a novel Parallel Dual-Path Update strategy to simultaneously refine object and pixel features.

We propose key improvements that drive superior performance. First, we remove the need for having a traditional two-stage model for the bounding box prediction process. Instead, we employ object queries guided by activation maps on training, allowing the model to focus on instance-specific features while maintaining high explainability. Secondly, we introduce a feature decoupling mechanism within each decoder layer to keep object and pixel-level features aligned, capturing better per-object semantic features. Lastly, we build on top of the classical U-Net architecture which

allows for sequential multi-scale feature propagation in our decoder. Our model shows on-par state-of-the-art performance across multiple diverse datasets while maintaining explainability and being robust.

The main contributions of this paper include:

1. We extend the U-Net architecture by integrating a query-based approach with a Transformer decoder, making the U-Net model instance aware.

2. We introduce a novel pixel decoder with decoupled mask and instance feature branching and a Parallel Dual-Path Update strategy within the Transformer decoder, which refines both object and pixel features simultaneously in U-Net's hierarchical fashion.

3. We employ object queries guided by activation maps during training, making our model explainable.

4. We introduce the novel 2025 Revvity Full Cell Segmentation Dataset, which comprises hundreds of images with thousands of manually annotated cell instances.

## 2 RELATED WORK

Mask R-CNN He et al. (2018) has set the standard for instance segmentation in natural images through its proposal-based approach. Building on Faster R-CNN Ren et al. (2016), Mask R-CNN adds a dedicated mask prediction branch, enabling end-to-end segmentation of individual instances. The process begins with detecting object bounding boxes, followed by applying Region of Interest (RoI) operations, such as RoI-Pooling Girshick et al. (2014) or RoI-Align He et al. (2018), to extract detailed region features for object classification and mask generation. While these two-stage, region-based methods have achieved high performance across various benchmarks, they are often hindered by inefficiencies from generating numerous redundant region proposals, limiting their scalability in practical, real-world applications.

The latest iteration, YOLOv8 Reis et al. (2024), represents a state-of-the-art solution for both object detection and instance segmentation, significantly improving COCO Mean Average Precision (mAP) scores. YOLOv8 introduces the C2f (Cross Stage Partial Fusion) building block, designed for more efficient feature extraction and fusion, enhancing both detection and segmentation tasks. Following this, YOLOv9 Wang et al. (2024) builds on YOLOv8 by introducing the GELAN (Gradient Enhanced Layer Aggregation Network) and PGI (Progressive Gradient Interpolation), which further enhance multi-scale feature fusion and improve the model's performance during training. In addition, the YOLO family employs an advanced data augmentation scheme, notably Mosaic Augmentation Hao & Zhili (2020), where images are transformed by stitching together four different images. This augmentation pushes the model to learn better generalization by exposing it to objects in diverse positions, levels of occlusion, and environments.

In biomedical image segmentation, where objects in microscopy typically have complex shapes, random orientations, and varying sizes, traditional axis-aligned bounding boxes perform poorly Follmann & König (2020), Kirillov et al. (2016). For instance, CellPose Stringer et al. (2021) provides a novel approach by generating topological maps through a simulated diffusion process. The method uses a U-Net architecture Ronneberger et al. (2015) to predict horizontal and vertical gradients, as well as a binary map of cell pixel predictions. These predicted gradients are then used to create a vector field that groups pixels by their directional flow towards the cell's center of mass. By tracking these gradients, CellPose successfully segments individual cells, although this method often requires an additional size model to predict object diameters and scale images, especially when faced with high variability in object sizes.

Query-based methods have gained prominence with the introduction of DETR Carion et al. (2020), which demonstrated the potential of a Transformer-based encoder-decoder architecture to achieve competitive results in detection and segmentation tasks. Unlike traditional region-based methods, query-based models rely on object queries to predict object instances directly, eliminating the need for handcrafted representations like bounding boxes. This shift marked a significant advancement in the efficiency and performance of instance segmentation models. Extensions such as Mask2Former and FastInst Cheng et al. (2022a) He et al. (2023) introduced masked attention for improved convergence and segmentation accuracy, while Mask DINO Li et al. (2022) unified object detection and

segmentation tasks into a single framework. Finally, U-Net has long been a standard for medical image segmentation, consistently demonstrating superior performance due to its use of skip connections and hierarchical decoder structures that capture rich contextual information. In this work, we introduce a query-based approach to a standard U-Net architecture, demonstrating that this adaptation significantly enhances instance segmentation performance in the medical domain

## 3 MODEL OVERVIEW

Instance segmentation is a critical task in computer vision, particularly for applications such as biomedical imaging, where identifying individual objects in complex environments is essential. Instance segmentation can be formulated as a task of grouping related pixels for each of the $N$ defined objects in an image. This process can may resemble clustering, where each object is represented as a cluster center, and the goal is to assign associated pixel features to their corresponding object. The object representation serves as the centroid, and pixels belonging to the same object are grouped together based on feature similarity. Recent works, such as DETR Carion et al. (2020) and Mask2Former Cheng et al. (2022a), have shown that a good instance representation is crucial in accurate segmentation tasks. Inspired by these models, we represent each object as a $D$-dimensional feature vector, forming instance embeddings also known as instance queries. These queries act as cluster centers in the $D$-dimensional feature space, guiding the assignment of pixel features to specific instances.

To effectively model both mask and instance features, we propose a convolutional Pixel decoder ?? with decoupled branches. One branch handles mask features, representing per-pixel embeddings of the entire image. The other branch models instance features and outputs a corresponding instance embeddings for guidance. Similar to a standard U-Net, our decoder incorporates skip connections to enrich semantic information from earlier layers, ensuring that both pixel and instance features benefit from multi-scale contextual information.

The Transformer decoder addresses the clustering idea by iteratively updating the mask and instance features in parallel and subsequently refining instance queries. Unlike traditional methods that perform multi-scale feature fusion before decoding, we utilize U-Net's hierarchical decoding structure, making the process iterative. In this approach, features from each decoder layer are passed sequentially to the next, allowing instance queries to be refined in a stepwise manner across multiple scales. The final instance mask predictions are decoded from the refined mask features and object queries.

## 4 PIXEL DECODER

Multi-scale and high-context features have proven to be crucial for segmentation tasks Chen et al. (2017) Wang et al. (2020b) Kirillov et al. (2019). In the biomedical domain, U-Net, with all its variants, still holds the ground as the most superior network for accurate segmentation.n. This is primarily due to the design of U-Net's decoder, which maintains high semantic consistency through the use of skip connections that transfer important features across layers.

We introduce a simple U-Net-like pixel decoder to propagate feature maps. Our pixel decoder works with three types of features: main features, mask features, and instance features 1. The main features serve a similar role to those in the vanilla U-Net, aggregating spatial context across the image. The instance and mask features, however, are specifically designed to support instance segmentation and are tightly integrated with the Transformer decoder. The mask features act as per-pixel embeddings, capturing rich semantic information, while the instance features are responsible for generating object queries at each level. Since both the mask and instance features are derived from the main feature map, they remain aligned, ensuring parallel information flow between pixel-level and object-level representations.

At each pixel decoder layer, given the main feature map $X$, we combine it with a skip connection from the encoder. The combined features are then passed through a simple double depth-wise convolution with residual connection to retain lightweight nature of pixel decoder. The result is a refined main feature map $X$, which we then use to decouple both mask features $X_m$ and instance features $X_i$.

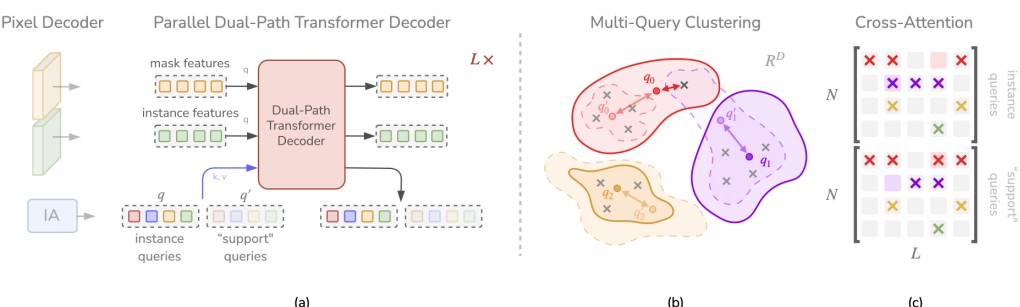

Figure 2: Overview of IAUNet's Parallel Dual-Path Transformer Decoder. (a) The Pixel Decoder generates mask and instance features, which are then processed by the Parallel Dual-Path Transformer Decoder. (b) Multi-query clustering assigns each object two queries, allowing for more robust feature representation and better captures complex shapes of objects. (c) Cross-attention is performed between instance queries and pixel features to refine object-level predictions – two attention matrices for $2N$ queries.

To maintain global consistency across layers, we process the main feature map $X$ separately with the upscaled mask features $X'_m$ and instance features $X'_i$ from the previous layer. Specifically, we concatenate $X$ with $X'_m$ to update the mask features, and $X$ with $X'_i$ to update the instance features. These concatenated features are processed by corresponding branches. We use two parallel stacked 3x3 convolution blocks for both the instance and mask branches. We use a simple bilinear interpolation is used to propagate all the features to the next decoder layer.

Unlike other methods that directly employ the feature maps from the pixel decoder to produce segmentation masks, we leverage a Transformer decoder to further refine these features. This design reduces the pixel decoder's need for heavy context aggregation and allows the Transformer decoder to handle the more complex instance segmentation refinement.

## 5 Guided Instance Queries

Central to this refinement process are guided instance queries, which ensure accurate object segmentation. Object queries play a crucial role in the Transformer decoder Liang et al. (2023) Carion et al. (2020). Since object queries are used to embed information about the object, they serve as the basis for accurate instance segmentation. Models like DETR Carion et al. (2020) and Mask2Former Cheng et al. (2022a) utilize either zero-initialized or learnable embeddings to describe instances without relying on prior knowledge of the image semantics.

In contrast, we introduce query guidance to avoid convergence into suboptimal local minima and to guide the model toward learning more informative object representations. At each level of the decoder, the model learns to generate guided queries, which capture denser and more accurate object representations. These instance embeddings get progressively refined through the decoder while preserving high-resolution object features.

At each decoder stage, the Instance Activation (IA) layer 1 generates $N$ guided instance queries $a \in \mathbb{R}^{N \times H \times W}$. Given the instance features $X_i$ from the Pixel decoder, the IA layer produces activation maps by highlighting important regions for each object. Formally, IA can be defined as:

$$a = \text{softmax}(f(X_i)) \in \mathbb{R}^{N \times H \times W} \tag{1}$$

where $f(x)$ is a simple 3x3 convolution followed by a softmax function to normalize the activations.

After obtaining normalized instance activation maps $a \in \mathbb{R}^{N \times H \times W}$, we select $N$ object queries from the instance features $X_i$ with high foreground probabilities from instance activations. We then perform an element-wise multiplication with the $X_i$ feature map to generate the final object queries: $q = a \cdot X_i^T \in \mathbb{R}^{N \times 256}$. Thus, each object gets encoded into a 256-dimensional vector.

The learning of instance activation maps is driven solely by how accurate the resulting instance predictions are. This eliminates the need for explicit guidance to optimize the activations. Since the model is guided only by the accuracy of the final segmentation, it can adapt its activations to represent highly variable object shapes without any rigid constraints.

# 6 PARALLEL DUAL-PATH TRANSFORMER DECODER

In the IAUNet model, we implement a Parallel Dual-Path Transformer Decoder that updates both object queries and pixel features in parallel. The key component of our Transformer decoder includes double-center clustering, where the object gets represented with two queries.

At each decoder layer $l$, we generate new instance queries $q$ from the instance features $X_i$ and concatenate them with $N$ instance queries from the previous layer (*"support"* queries) to obtain a total of $2N$ instance queries 2. Each object is represented with two queries *("two cluster centers")*. The total $2N$ object queries, $q \in \mathbb{R}^{2N \times 256}$, are processed with the flattened high-resolution mask and instance features $X_m \in \mathbb{R}^{L \times 256}$ and $X_i \in \mathbb{R}^{L \times 256}$, where $L = H_l \times W_l$ for the $l$-th decoder layer.

The Parallel Dual-Path Transformer performs parallel mask and instance features update and query update. The new instance queries hold rich object features and act as primary cluster centers. While the previous instance queries function as support centers. Such dual representation allows the model to better capture complex object structures by associating pixel features with two distinct queries.

## 6.1 POSITIONAL EMBEDDINGS

To maintain spatial awareness, which is crucial for Transformer-based models, we add learnable positional embeddings to object queries. For the *"support"* queries, we use additional N learnable positional embeddings. For each resolution, we add sinusoidal positional embeddings $e_{pos} \in \mathbb{R}^{H_l W_l \times D}$ to the mask and instance features $X_m$ and $X_i$ following [ref].

## 6.2 PIXEL FEATURES UPDATE

We refine both the mask and instance pixel features in parallel. Since mask features $X_m$ are crucial for describing the semantics of the entire image, the model learns to associate such features with individual objects. Instance features $X_i$, on the other hand, are the key to predicting correct activation maps. In the parallel feature update, we first want to associate each object with its set of pixel features. For each mask and instance features we use cross-attention layers followed by a feed-forward network (FFN):

$$X_l = \text{softmax}\left(Q_l K_l^T\right) V_l + X_{l-1} \tag{2}$$

Here $Q_l \in \mathbb{R}^{H_l W_l \times 256}$ are the pixel features at the $l$-th layer and $K_l, V_l \in \mathbb{R}^{2N \times 256}$ refer to $2N$ $D$-dimensional instance features $q_l$. For each set of pixel features $X_m$, $X_i$ and the $2N$ object queries $q$, the attention matrix $M \in \mathbb{R}^{L \times 2N}$ can be intuitively divided into two subgroups. The first group captures the attention of the new instance queries toward the pixel features, while the second group focuses on the support queries. Both object queries come from the relatively same region of features guided by learnable activation maps. Therefore, the attention matrices 2 between pixel features and queries for both groups are expected to be quite similar. The support queries are meant to match the correct pixel features back to the instance cluster, even if the new instance queries have less attention to these pixel features. The whole process tries to make the feature-query update smoother by accounting for object information from previous layers.

Finally, the newly refined mask and instance features are passed to the next decoder level, ensuring consistent multi-scale updates across layers.

## 6.3 INSTANCE QUERIES UPDATE

Assymetrically, we update $2N$ instance queries with respect to the instance features $X_i$. We use cross-attention layer followed by the self-attention layer and FFN layer. This design maintains awareness between all the queries ensuring full object separation.

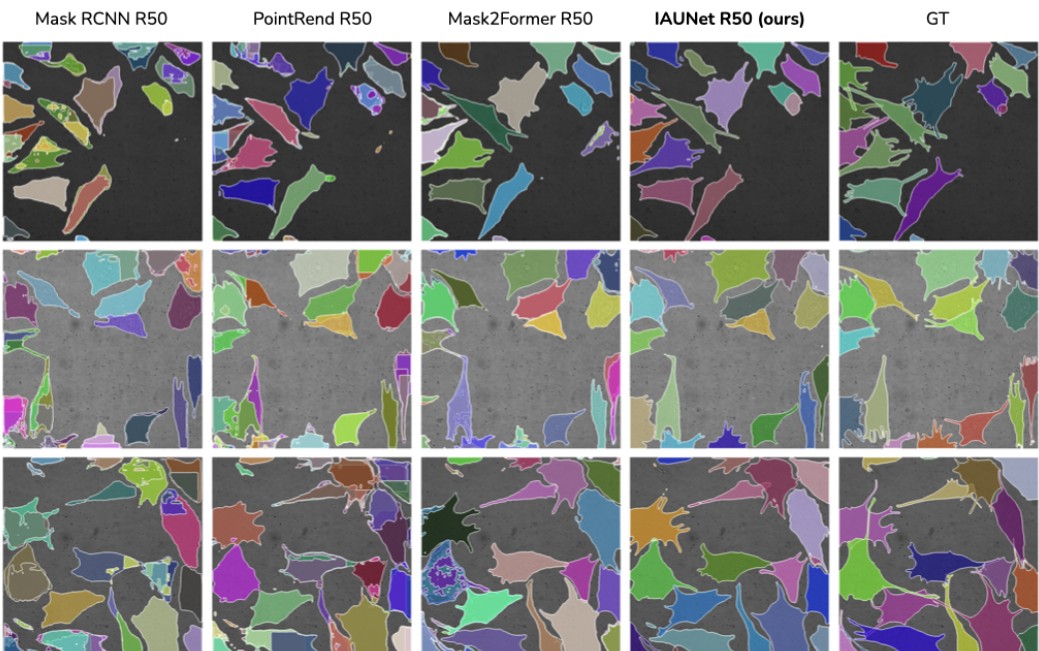

Figure 3: Comparison of instance segmentation performance between Mask R-CNN, PointRend Kirillov et al. (2020), Mask2Former, and IAUNet models with a ResNet-50 backbone on the Revvity-25 dataset.

## 7 MASK LEVEL MATCHING

During training the model outputs $N$ instance mask predictions. To supervise the model's training, we utilize a matching strategy to assign predictions to the gt masks and compute losses. We employ the optimal bipartite matching Carion et al. (2020) Cheng et al. (2022b), resulting in a set of corresponding {*prediction, ground-truth*} instance mask pairs. We adopt one-to-one label assignments to get the best predictions. Given a set of $M$ ground truth masks $G = \{g_0, g_1, \ldots, g_m\}$ and a fixed-size set of $N$ predictions $P = \{p_0, p_1, \ldots, p_n\}$, where $N > M$, we calculate losses in the subset of best-matched predictions of $P$. The one-to-one matching assignment finds a minimum weighted bipartite graph matching $\sigma \in S$ within the sets $G$ and $P$:

$$\sigma = \arg \min_{\sigma \in S} \sum_{i=1}^{n} C(p_{\sigma(i)}, g_i) \tag{3}$$

where $\sigma$ is the permutation representing the matching between predicted and ground truth masks that minimizes the sum, $S$ is the set of permutations, and $C$ is a pair-wise matching cost between $G$ and $P$ that is a weighted combination of both classification cost $C_{cls}$ and mask regression cost $C_{mask} = \{C_{dice}, C_{bce}\}$. Each target is assigned to an object prediction through an optimal assignment problem computed efficiently using the Hungarian algorithm **?**. With the Hungarian approach, we find the optimal match between $M$ ground truth objects and $N$ predictions given a weighted cost matrix $C$

We define the matching cost functions in alignment with the calculation of our losses to maintain consistency. The weights assigned to all the cost functions correspond to the weights applied to all the losses. Specifically, we set the coefficient $\lambda_{cls}$ to 1.0, $\lambda_{dice}$ to 2.0, and $\lambda_{bce}$ to 5.0.

$$C = C_{cls} \cdot \lambda_{cls} + C_{dice} \cdot \lambda_{dice} + C_{bce} \cdot \lambda_{bce} \tag{4}$$

During inference, we re-score the predicted masks and use non-maximum suppression (NMS). We leverage the classification scores to assess the confidence level of each predicted instance. Simultaneously, for each instance we calculate the maskness metrics Wang et al. (2020a), denoted as

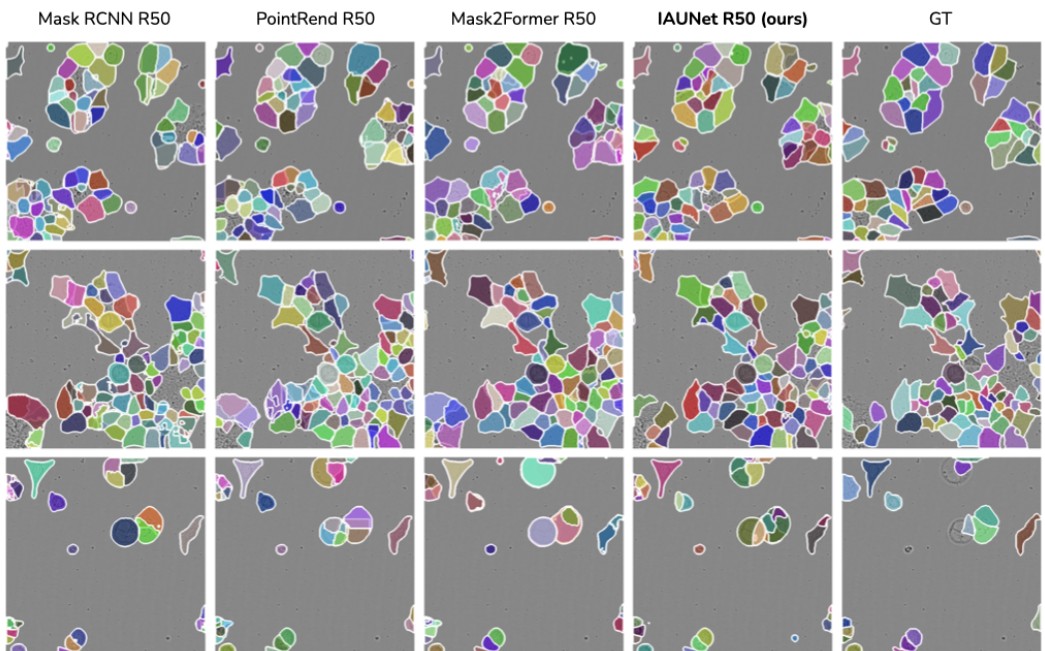

| Mask RCNN R50 | PointRend R50 | Mask2Former R50 | **IAUNet R50 (ours)** | GT |

Figure 4: Instance segmentation performance comparison of Mask R-CNN, PointRend, Mask2Former, and IAUNet models with a ResNet-50 backbone, evaluated on the LiveCell dataset.

$m_i = \frac{1}{N} \sum_{i=1}^{N} p_i$, where $p$ is the predicted probability mask with $N$ pixels. Thus, the combined confidence score $s$ is computed as a contribution of both class confidence $c$ and maskness scores $m$:

$$s_i = c_i \cdot m_i$$

## 8 EXPERIMENTS

**Datasets.**

We evaluate our performance on several datasets, including our novel Revvity-25 dataset. We report Average Precision scores for the LiveCell, EVICAN, and NeurIPS-CellSeg 22 datasets. For each dataset, we preprocess all images to contain a maximum of 100 instances.

**Evaluation Metrics.** For our main results of instance segmentation, we report COCO **?** mask AP scores on the test subsets for all datasets. We specifically focus on the AP to get a general understanding of model's performance. We also propose to compare the performance with the state-of-the-art models in both natural and cellular domains.

### 8.1 TRAINING SETTINGS

All experiments were conducted on a single Tesla V100 GPU with 32GB of memory. Our model is implemented using the PyTorch framework (torch==2.3.1) Paszke et al. (2019) and runs on CUDA 12.1. We adopt the training scheme published in earlier works Cheng et al. (2022a). We use the CosineAnnealingLR scheduler Loshchilov & Hutter (2017) with a minimum learning rate of 1e-6, and the AdamW optimizer Loshchilov & Hutter (2019) with an initial learning rate of 1e-4 and a weight decay of 0.05. During training, we employ longest-side resizing to scale all images to 512x512 pixels while maintaining their original aspect ratio. For data augmentation, we adopt scale jittering augmentation Cheng et al. (2022b) with a random scale sampled from the range 0.8 to 1.5 followed by a fixed size crop to 512x512 and random flipping. We follow a consistent augmentation strategy across all models and benchmarks. All models were trained until full convergence with a batch size of 8. Unless specified, we use the same longest-side resizing processing to test and benchmark models. During inference, we maintained the same thresholds for the non-maximum

Table 1: Instance segmentation performance comparison of various models on multiple datasets, including LiveCell-Crop, NeurIPS22-CellSeg challenge, and EVICAN2, with different backbone architectures. We differentiate all model architectures by their subgroup convolutional and transformer-based backbones as well as YOLO and SAM Kirillov et al. (2023) family models and CellPose model with additional Size Model (SM) Stringer et al. (2021).

| Models | backbones | LiveCell AP | LiveCell $AP_{50}$ | NeurIPS22 AP | NeurIPS22 $AP_{50}$ | $EVICAN2_E$ AP | $EVICAN2_E$ $AP_{50}$ | $EVICAN2_M$ AP | $EVICAN2_M$ $AP_{50}$ | $EVICAN2_D$ AP | $EVICAN2_D$ $AP_{50}$ | #params. | FLOPs |
|---|---|---|---|---|---|---|---|---|---|---|---|---|---|
| **Models with Convolution-Based Backbones** | | | | | | | | | | | | | |
| Mask R-CNN | R50 | **44.7** | **74.2** | 52.8 | 74.7 | 48.1 | 75.9 | 20.7 | 42.5 | 19.1 | 39.8 | 44M | 115G |
| PointRend | R50 | 44.0 | 73.5 | **54.7** | **74.8** | 26.6 | 47.9 | 18.0 | 38.5 | 13.4 | 28.3 | 56M | 66.3G |
| Mask2Former | R50 | 43.7 | 73.8 | 42.9 | 66.6 | **53.4** | **89.1** | 29.1 | 54.9 | 24.2 | 50.4 | 44M | 66.2G |
| **IAUNet (ours)** | R50 | **44.7** | 73.9 | 49.0 | **75.1** | 53.3 | 85.6 | **29.2** | **55.0** | **25.3** | 47.9 | 65M | 292.6G |
| Mask R-CNN | R101 | 44.2 | 73.2 | 53.3 | 73.2 | 41.5 | 69.9 | 23.3 | 46.9 | 17.8 | 36.7 | 63M | 134G |
| PointRend | R101 | 44.0 | 73.7 | 52.0 | **76.0** | 41.3 | 65.2 | 20.2 | 39.3 | 14.8 | 32.1 | 75M | 85.7G |
| Mask2Former | R101 | 44.0 | 73.5 | 44.2 | 68.3 | 54.4 | 87.8 | 27.1 | 51.7 | 20.4 | 42.4 | 63M | 85.6G |
| **IAUNet (ours)** | R101 | **44.7** | **74.1** | 49.3 | 74.6 | **59.6** | **88.7** | **29.8** | **52.9** | **28.5** | **52.6** | 84M | 331.6G |
| **Models with Transformer-Based Backbones** | | | | | | | | | | | | | |
| Mask R-CNN | Swin-S | 44.3 | 73.3 | **55.4** | 76.2 | - | - | - | - | - | - | 69M | 141G |
| PointRend | Swin-S | 43.9 | 73.5 | 54.6 | 76.5 | - | - | - | - | - | - | 81M | 92.9G |
| Mask2Former | Swin-S | **44.6** | **74.3** | 43.9 | 67.9 | - | - | - | - | - | - | 69M | 92.8G |
| **IAUNet (ours)** | Swin-S | 43.9 | 73.6 | 52.4 | 72.8 | - | - | - | - | - | - | 77M | 328G |
| Mask R-CNN | Swin-B | 44.2 | 73.1 | **56.0** | 76.6 | - | - | - | - | - | - | 107M | 179G |
| PointRend | Swin-B | 44.0 | 73.7 | 55.0 | **77.1** | - | - | - | - | - | - | 119M | 131G |
| Mask2Former | Swin-B | **44.9** | **74.7** | 46.3 | 70.9 | - | - | - | - | - | - | 107M | 134G |
| **IAUNet (ours)** | Swin-B | 44.0 | 73.4 | 55.8 | **80.3** | - | - | - | - | - | - | 117M | 412G |
| **YOLO Family** | | | | | | | | | | | | | |
| YOLOv8-M | - | 37.5 | 72.2 | 44.9 | 81.1 | 43.8 | 82.3 | 27.5 | 57.1 | 20.0 | 46.2 | 27.2M | 110.4G |
| YOLOv8-L | - | 40.5 | 72.5 | 45.4 | 81.5 | 44.7 | 83.1 | 28.1 | 58.2 | 20.3 | 46.1 | 45.9M | 220.8G |
| YOLOv8-X | - | 41.1 | 73.1 | 47.7 | 81.4 | **45.8** | **85.6** | 28.9 | 59.2 | 20.7 | **47.3** | 71.8M | 344.5G |
| YOLOv9-C | - | 41.2 | 73.2 | 46.9 | 81.6 | 45.6 | 84.4 | 27.2 | 57.9 | 20.1 | 47.3 | 27.8M | 159.1G |
| YOLOv9-E | - | 41.4 | 73.1 | **47.6** | **82.8** | 45.9 | **85.6** | **28.3** | **59.8** | 22.2 | 49.9 | 60.5M | 248.1G |
| **IAUNet (ours)** | R50 | **44.7** | **73.9** | 49.0 | 75.1 | 53.3 | 85.6 | 29.2 | 55.0 | 25.3 | 47.9 | 65M | 292.6G |
| **CellPose Family** | | | | | | | | | | | | | |
| CellPose | - | 34.5 | 60.1 | 32.9 | 51.5 | 0.9 | 2.8 | 0.1 | 0.3 | 0.0 | 0.0 | 6.6M | 163.6G |
| CellPose + *SM* | - | 34.9 | 60.4 | 44.1 | 74.8 | 8.7 | 16.8 | 1.6 | 4.4 | 2.3 | 6.8 | 6.6M | 163.6G |
| **IAUNet (ours)** | R50 | **44.7** | **73.9** | **49.0** | **75.1** | **53.3** | **85.6** | **29.2** | **55.0** | **25.3** | **47.9** | 65M | 292.6G |
| **SAM Family** | | | | | | | | | | | | | |
| SAM-B *(points)* | - | 5.0 | 12.4 | 30.7 | 56.6 | 28.4 | 56.0 | 5.4 | 13.8 | 3.2 | 7.2 | 90M | 742G |
| SAM-B *(boxes)* | - | 24.3 | 56.9 | **54.3** | **91.7** | 55.0 | **96.6** | **38.6** | 91.2 | **34.8** | **82.3** | 90M | 742G |
| **IAUNet (ours)** | R101 | **44.7** | **74.1** | 49.3 | 74.6 | **59.6** | 88.7 | 29.8 | 52.9 | 28.5 | 52.6 | 84M | 331.6G |

suppression overlap and confidence for objects and used the same mask prediction threshold of 0.5 for all the trained models.

## 8.2 RESULTS

In Table 1, we compare the performance of IAUNet with other state-of-the-art models such as Mask R-CNN, PointRend, and Mask2Former across several datasets, including LiveCell, NeurIPS22, and EVICAN2. For models utilizing the ResNet-50 backbone, IAUNet shows competitive performance, especially on the EVICAN2 datasets. On the $EVICAN2_{Easy}$ dataset, IAUNet achieves an AP of 53.3, which is marginally lower than the 53.4 obtained by Mask2Former but significantly higher than both Mask R-CNN (48.1) and PointRend (26.6). Notably, IAUNet achieves superior $AP_{50}$ on the same dataset, with 85.6, second only to Mask2Former (89.1). On the $EVICAN2_{Medium}$ dataset, IAUNet outperforms all other models in both AP (29.2) and $AP_{50}$ (55.0), indicating its strong ability to segment verying in size objects in complex scenes. On the LiveCell dataset, IAUNet achieves an AP of 44.7, with similar performance to Mask R-CNN but surpassing PointRend (44.0) and Mask2Former (43.7). Across the YOLO family of models, IAUNet demonstrates significant performance improvements on the YOLOv8 and YOLOv9 models.

We perform an evaluation of the IAUNet model, comparing it with popular state-of-the-art instance segmentation models such as Mask R-CNN, PointRend, and Mask2Former on our Revvity-25. The dataset offers a challenging benchmark for instance segmentation tasks due to the complex shapes and varying sizes of cells.

| | | | | | | | | | |
|---|---|---|---|---|---|---|---|---|---|
| | | | | *Revvity-25* | | | | | |
| Models | backbone | AP | $AP_{50}$ | $AP_{75}$ | $AP_S$ | $AP_M$ | $AP_L$ | #params. | FLOPs |
| *Models with Convolution-Based Backbones* | | | | | | | | | |
| Mask R-CNN | R50 | 40.8 | 79.8 | 38.4 | 0.1 | 20.7 | 45.4 | 44M | 115G |
| PointRend | R50 | 45.1 | 83.2 | 47.0 | 0.1 | 25.1 | 50.0 | 57M | 66.3G |
| Mask2Former | R50 | 40.2 | 73.0 | 41.7 | **0.8** | 16.6 | 46.2 | 44M | 66.2G |
| **IAUNet** (ours) | R50 | **51.4** | **84.6** | **55.9** | 0.5 | **27.7** | **58.0** | 65M | 292.6G |
| Mask R-CNN | R101 | 39.0 | 79.1 | 35.5 | 0.4 | 18.7 | 43.4 | 63M | 134G |
| PointRend | R101 | 44.4 | 82.4 | 44.5 | 0.0 | 20.7 | 49.6 | 75M | 85.7G |
| Mask2Former | R101 | 44.4 | 78.4 | 46.7 | 0.9 | 20.7 | 50.6 | 63M | 85.6G |
| **IAUNet** (ours) | R101 | **51.0** | **83.0** | **55.7** | **1.5** | **28.0** | **57.8** | 84M | 331.6G |
| *Models with Transformer-Based Backbones* | | | | | | | | | |
| Mask R-CNN | Swin-S | 24.1 | 59.2 | 14.4 | 0.0 | 6.6 | 28.1 | 69M | 141G |
| PointRend | Swin-S | 48.0 | 85.8 | 51.1 | 0.4 | 25.3 | 53.3 | 81M | 92.9G |
| Mask2Former | Swin-S | 37.6 | 65.6 | 40.1 | 0.1 | 16.3 | 43.7 | 69M | 92.8G |
| **IAUNet** (ours) | Swin-S | **53.3** | **86.2** | **58.3** | **1.8** | **29.9** | **59.8** | 77M | 328G |
| Mask R-CNN | Swin-B | 18.8 | 50.6 | 8.4 | 0.0 | 3.7 | 22.5 | 107M | 179G |
| PointRend | Swin-B | 45.9 | 83.2 | 46.2 | 0.1 | 24.4 | 51.0 | 119M | 131G |
| Mask2Former | Swin-B | 52.0 | 84.1 | 57.4 | 1.0 | 28.1 | 58.7 | 107M | 134G |
| **IAUNet** (ours) | Swin-B | **52.8** | **85.0** | **58.7** | 1.2 | **29.7** | **59.2** | 117M | 412G |

Table 2: Performance comparison of instance segmentation models with ResNet-50, ResNet-101, Swin-S, and Swin-B backbones on the Revvity-25 dataset.

**Convolution-Based Backbones** In 2 models using the ResNet-50 backbone, IAUNet achieves an **AP** of **51.4**, outperforming PointRend (**45.1**) and Mask2Former (**40.2**). IAUNet also achieves the highest $AP_{50}$ (84.6) and $AP_{75}$ (55.9), showcasing its strong performance in detecting and segmenting instances at varying IoU thresholds. IAUNet shows particular strength in medium and large object detection, achieving 27.7 in $AP_M$ and 58.0 in $AP_L$, both higher than its competitors.

With the ResNet-101 backbone, IAUNet maintains its lead, scoring **51.0** in **AP**, while PointRend and Mask2Former hover around **44.4**. The improvement is more prominent in the segmentation of medium and large objects, further confirming the model's ability to handle complex object structures better than traditional region-based approaches.

**Transformer-Based Backbones** With Swin-S and Swin-B backbones, IAUNet further extends its performance lead, achieving **53.3** and **52.8** in **AP**, respectively. In comparison, PointRend reaches 48.0 and 45.9, while Mask2Former achieves 52.0 on **Swin-B** but struggles on smaller object instances. IAUNet demonstrates superior segmentation of medium and large objects, achieving 29.7 and 59.2 on Swin-B, highlighting its ability to handle objects of varying sizes 3 without relying on bounding box detections that lead to duplicate proposals.

## 9 LIMITATIONS AND CONCLUSION

In this work, we introduced IAUNet, a novel architecture combining U-Net with query-based mechanisms for instance segmentation. The model's Instance Activation layers generate guided object queries, while the Parallel Dual-Path Transformer Decoder refines features across multiple scales. IAUNet outperforms leading models, especially in handling medium and large objects, and sets a new baseline for biomedical imaging tasks, as demonstrated on the 2025 Revvity Full Cell Segmentation Dataset.

IAUNet faces challenges with small object segmentation, similar to other query-based methods Cheng et al. (2022a); He et al. (2023). Additionally, IAUNet could be optimized to handle a higher number of instances per image. Future research should focus on developing more efficient solutions for small object segmentation.

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
