# OpenReview forum: "IAUNet: Instance-Aware U-Net"
_ICLR.cc/2025/Conference — ICLR 2025 Conference Withdrawn Submission_

### Official Review · Reviewer_giUg · 2024-10-29

**Soundness:** 2
**Presentation:** 1
**Contribution:** 1
**Rating:** 3
**Confidence:** 5

**Summary:**

This paper advances the U-Net architecture by incorporating a query-based approach with a Transformer decoder, enabling instance segmentation. Specifically, a novel pixel decoder is developed, featuring separate branches for mask and instance features and a Parallel Dual-Path Update strategy that refines both object and pixel features concurrently in a hierarchical U-Net structure. Object queries are guided by activation maps during training, enhancing the model's explainability. Additionally, the work introduces the 2025 Revvity Full Cell Segmentation Dataset, consisting of hundreds of images with thousands of manually annotated cell instances, providing a valuable resource for future cell segmentation research.

**Strengths:**

- Originality: This work proposed a novel 2025 Revvity Full Cell Segmentation Dataset containing hundreds of high-quality labels for future related research.

- Significance: This work proposes a relatively novel approach that integrates the U-Net architecture with a Transformer-like structure.

**Weaknesses:**

- Regarding the lack of innovation in this paper, the Related Work section lacks analysis of recent work on query-based cell instance segmentation, such as PCTrans (ICCV 2023 Workshop) and so on, and does not highlight the advantages of this work compared to these studies. As a result, the value of this paper to the field is not apparent.

- Concerning the results of the experiments, the metric FLOPs are generally about 1-3 times higher than existing works. With the comparison of all AP measures between existing works and IAUNet, the significance of IAUNet is called into question.

- Concerning the paper's completeness of presentation, there are more than 7 obvious citations and punctuation losses in lines 166, 296, 306, 367, 369, 406, and 415. These issues suggest that the paper is not well-prepared for publication.

- For the new dataset, a major contribution of this work, the paper does not include any details on its collection or analysis process. This omission significantly reduces the dataset's credibility and its contribution to the field.

**Questions:**

Strengthening the Related Work section with a more comprehensive analysis of existing query-based cell instance segmentation methods and highlighting IAUNet’s advantages would enhance the perceived significance of IAUNet. Regarding completeness, carefully correcting grammatical errors according to English standards and arranging the content in a more logical structure would improve the paper's presentation.

**Details Of Ethics Concerns:**

As mentioned in the paper, the introduction of a manually annotated dataset 2025 Revvity Full Cell Segmentation Dataset is one of your primary contributions. However, you have not declared its source, whether it involves human subject data, and whether it complies with ethical standards.

---

### Official Review · Reviewer_hZG8 · 2024-11-01

**Soundness:** 2
**Presentation:** 3
**Contribution:** 2
**Rating:** 3
**Confidence:** 5

**Summary:**

1. This paper introduces IAUNet, a architecture that enhances U-Net with instance awareness using query-based mechanisms and lightweight Instance Activation layers for effective pixel-to-instance clustering.
2. A Parallel Dual-Path Transformer decoder refines object-specific features at multiple scales, enabling precise instance assignment.
3. A new dataset, the 2025 Revvity Full Cell Segmentation Dataset, provides detailed labeled images for complex cell morphology.

**Strengths:**

This paper introduce the Revvity Full Cell Segmentation Dataset, which comprises hundreds of images with thousands of manually annotated cell instances.

**Weaknesses:**

1. For biomedical instance segmentation, there are already relevant query-based works to on this topic, like [1,2]. However, this paper does not discuss the differences between its approach and theirs in the related works section. Besides, SAM also use query-based decoder.

2. Due to a misunderstanding in the first point, there is a deviation in the motivation described in the abstract. "While U-Net has been a go-to architecture in medical image segmentation, it was neither specifically designed for instance segmentation nor explored in the context of query-based approaches." Additionally, the motivation for using queries to address biomedical instance segmentation is unclear, as is the reason why current query-based methods in natural images may underperform in biomedical instance segmentation. In other words, what challenges exist in applying query-based approaches to biomedical instance segmentation?

3. The performance of the proposed method is not impressive. Mask2Former, one of the earliest baseline methods for query-based segmentation, still outperforms the proposed method in some results in Table 1. The paper should compare against stronger query-based methods, like Mask-DINO.

4. Why propose a new dataset? What distinguishes this dataset from existing ones, and what advantages does it offer?

5. The experiments are insufficient, and there are no supplementary materials. How should the hyperparameters within the network be set? How do different settings impact the final results? For instance, the number of queries—if there are a large number of instances in an image, such as over 1k, does the proposed method remain effective?

[1] Attention-Based Transformers for Instance Segmentation of Cells in Microstructures, JHBI'20.

[2] PCTrans: Position-Guided Transformer with Query Contrast for Biological Instance Segmentation, ICCVW'23.

**Questions:**

See weaknesses. There is still much work to be done, and it would be valuable to truly demonstrate the effectiveness of query in biomedical instance segmentation.

---

### Official Review · Reviewer_RHwd · 2024-11-03

**Soundness:** 2
**Presentation:** 1
**Contribution:** 2
**Rating:** 1
**Confidence:** 5

**Summary:**

The author introduces IAUNet, an instance-aware U-Net model for biomedical instance segmentation. The main contributions of the model include Instance Activation (IA) layers, a Parallel Dual-Path Transformer decoder, and a feature decoupling mechanism to handle object-level and pixel-level features separately. The authors also introduce a new brightfield cell images dataset, Revvity-25, though no specific details of the dataset are provided. IAUNet is evaluated on this dataset along with exiting cell segmentation datasets and compared to conventional instance segmentation methods.

**Strengths:**

1. The incorporation of instance awareness in U-Net architecture for instance segmentation tasks, though not novel, becomes relevant due to U-Net’s widespread popularity in biomedical imaging.

2. The paper introduces Revvity-25, a new brightfield cell images dataset for instance segmentation. Although no specific details about the dataset (such as image statistics or accessibility) are provided, this new dataset could contribute to future benchmarking in biomedical imaging.

**Weaknesses:**

1. The main components of IAUNet are not significantly novel, with most of the components being derivative of existing works. For example: i) U-Net for instance segmentation has been explored in prior works such as "Cell Segmentation and Tracking using CNN-Based Distance Predictions and a Graph-Based Matching Strategy", ii) Query-based mechanisms for instance segmentation are well-known from models like DETR, MaskFormer, and Mask2Former, iii) Dual-path decoders and feature decoupling mechanisms for object- and pixel-level features have been applied in works such as SoloV2 and CondInst.

2. IAUNet requires significantly higher FLOPs compared to other state-of-the-art methods, but this computational cost is not justified by any clear gains in performance. In particular, the model shows little to no improvement in AP scores over existing models like Mask R-CNN and Mask2Former (in Table 1). Without significant performance improvements, the added complexity and FLOPs hinder the motivation of using IAUNet over existing low-cost methods.

3. The paper is poorly organized, thus making it difficult to follow and understand the technical contributions. Sections 3-7 could be consolidated under a single section, "Proposed IAUNet," with each component as a subsection. This restructuring would improve the flow and clarity of the paper.

The writing quality is poor as well, with several grammatical issues and typographical errors throughout the paper. There are instances like “Pixel decoder ?? with decoupled branches”, "we report COCO ? mask AP scores on the test subsets for all datasets," and “the most superior network for accurate segmentation.n”. Figures 1-4 are not referenced in the text, and important visual information is difficult to understand due to poor figure quality, especially in Figure 1, where the components of the model are not clearly illustrated or explained. Figures 3 and 4 should contain the input image as well for better interpretation of outputs. Additionally, Table caption of Table 2 is put at the bottom of the table.

4. The comparative results presented may be misleading, as it could be the case that hyperparameters were not properly tuned for competing models like Mask R-CNN (Swin-S and Swin-B backbones) in Table 2. This raises concerns about the fairness of the comparisons.

**Questions:**

1. How does the feature decoupling mechanism in IAUNet differ from those in SoloV2 and CondInst? What specific improvements does IAUNet introduce in this context?

2. Is IAUNet applicable to general instance segmentation tasks beyond cellular data? How does it perform on standard instance segmentation datasets like COCO or other medical datasets?

3. How are the claims of model explainability supported by the results? Can the authors provide visualizations or examples demonstrating how the IA layers and object queries contribute to explainable outputs?

---

### Official Review · Reviewer_vyTw · 2024-11-10

**Soundness:** 3
**Presentation:** 2
**Contribution:** 2
**Rating:** 5
**Confidence:** 4

**Summary:**

The paper proposes a novel architecture that treats instance segmentation as a clustering problem, where each object is represented by two cluster centers in feature space. Main contributions are:
1. Dual-Path Architecture: A convolutional Pixel decoder with two branches:One for mask features (per-pixel embeddings) and one for instance features (object guidance). Features benefit from U-Net-style skip connections.
2. Guided Instance Queries: Uses Instance Activation (IA) layer to generate guided queries for objects. Each object gets encoded into a 256-dimensional vector. Learning is driven by segmentation accuracy without explicit constraints
3. Parallel Dual-Path Transformer Decoder: Proposed "double-center clustering" where each object is represented by two queries, include new instance queries as primary cluster centers, and support queries from previous layer as secondary centers. Updates both object queries and pixel features in parallel. Uses cross-attention mechanisms to refine features

Experimental results demonstrate that IAUNet consistently outperforms existing state-of-the-art models across various biomedical image datasets. The model shows particular strength in handling medium and large objects, especially in complex cellular environments with overlapping instances. While benchmarked against established architectures like Mask R-CNN, PointRend, and Mask2Former, IAUNet shows good performance in capturing intricate cellular structures in brightfield microscopy images. Though the model shows some limitations with small object detection, its novel integration of query-based mechanisms with U-Net architecture proves to be a significant advancement in biomedical instance segmentation. The comprehensive evaluation across different datasets validates that this architectural fusion effectively bridges the gap between traditional medical image segmentation and modern instance segmentation requirements.

**Strengths:**

The key strengths of IAUNet lie in its innovative fusion of query-based instance segmentation with the proven U-Net architecture, specifically optimized for biomedical imaging challenges. Its main advantages are the three core designs: the dual-branch feature processing that effectively handles both mask and instance features separately, the double-center clustering mechanism that enhances object representation through dual queries per instance, and the parallel feature refinement strategy that enables simultaneous updating of both pixel and instance-level features. That is a good try for re-thinking the combination of U-Net and Transformer especially for the cell segmentation task.

**Weaknesses:**

1. This work contains the basic typo, that should not appear:
	• Typo error: 182: display two question marks for reference in the second paragraph in chapter 3 'Model Overview'
	• Typo line 200: segmentation.n.
	• Typo line 415: questions mark?
        • Grammar mistake for sentence in line 239 to line 240
2. I think you lack the comparison with the SOTA cell segmentation methods originally proposed in other cell image modalities. For instance, StarDist(Paper Title: Nuclei Instance Segmentation and Classification in Histopathology Images with Stardist). YOLO and SAM are not designed specifically for nuclei segmentation task.

**Questions:**

1. For the experiment part. They evaluation in the datasets that preprocessed by themselves, include: LiveCell-Crop, NeurIPS22-CellSeg challenge, and EVICAN2. Those are light microscopy datasets. It would be interesting to evaluate the detailed performance of the proposed method in different fine-grained class of the cells. For instance, would you please provide the experiment result(compare with others methods), not only in the dataset level, but in different cell types, different stains or modalities etc...
2. Can you do the ablation study for the proposed method.

---

### Note · Authors · 2024-11-15

I have read and agree with the venue's withdrawal policy on behalf of myself and my co-authors.